# Network Pharmacology and Experimental Validation to Investigate the Antidepressant Potential of *Atractylodes lancea* (Thunb.) DC.

**DOI:** 10.3390/life12111925

**Published:** 2022-11-18

**Authors:** Ly Thi Huong Nguyen, Nhi Phuc Khanh Nguyen, Khoa Nguyen Tran, Heung-Mook Shin, In-Jun Yang

**Affiliations:** Department of Physiology, College of Korean Medicine, Dongguk University, Gyeongju 38066, Republic of Korea

**Keywords:** *Atractylodes lancea* (Thunb.) DC., depression, network pharmacology, molecular docking, tail suspension test

## Abstract

*Atractylodes lancea* (Thunb.) DC. (AL) has been indicated in traditional prescriptions for the treatment of depression. However, the mechanism of action of AL in the treatment of depression is still unclear. This study aimed to investigate the antidepressant potential of AL using network pharmacology, molecular docking, and animal experiments. The active components of AL were retrieved from the traditional Chinese medicine systems pharmacology database and analysis platform (TCMSP), and the depression-related targets were screened through the DisGeNET database. Overlapping targets of AL and depression were selected and analyzed. Ten active compounds of AL showed anti-depressant potential, including stigmasterol, 3β-acetoxyatractylone, wogonin, β-sitosterol, selina-4(14),7(11)-dien-8-one, atractylenolide I, atractylenolide II, atractylenolide III, patchoulene, and cyperene. These compounds target 28 potential antidepressant genes/proteins. Gene Ontology (GO) enrichment analysis revealed that the potential targets might directly influence neural cells and regulate neuroinflammation and neurotransmitter-related processes. The potential Kyoto Encyclopedia Genes and Genomes (KEGG) pathways for the antidepressant effects of AL include neuroactive ligand–receptor interactions, calcium signaling pathways, dopaminergic synapse, interleukin (IL)-17 signaling pathways, and the pathways of neurodegeneration. IL-6, nitric oxide synthase 3 (NOS), solute carrier family 6 member 4 (SLC6A4), estrogen receptor (ESR1), and tumor necrosis factor (TNF) were the most important proteins in the protein–protein interaction network and these proteins showed high binding affinities with the corresponding AL compounds. AL showed an antidepressant effect in mice by decreasing immobility time in the tail suspension test and increasing the total contact number in the social interaction test. This study demonstrated the antidepressant potential of AL, which provides evidence for pursuing further studies to develop a novel antidepressant.

## 1. Introduction

Depression is one of the most common mental disorders that negatively affect the quality of life of patients. The prevalence of depression was approximately 3% of the world’s population or 246 million people in 2020 [1]. Depression is characterized by emotional and behavioral symptoms, including low mood, loss of interest, the feeling of worthlessness, and suicidal thoughts [2]. Depression is a comorbid condition commonly observed in patients with chronic diseases, such as cancer, diabetes, and cardiovascular diseases [3]. Moreover, depression is also associated with other psychiatric disorders, such as anxiety and alcohol use disorder [4,5]. The link between depression and other diseases increases the risk of recurrence with higher severity and mortality rates, and increases the cost and duration of treatment, leading to an economic burden [6]. However, currently available drugs for the treatment of depression have limited efficacy and are associated with significant side effects [7,8]. This has raised the need to investigate and develop alternative antidepressants with higher effectiveness and fewer side effects.

Traditional herbal medicines have been considered an effective alternative therapy for the treatment of depression. Additionally, combinations of herbal and western medicine have exhibited superior therapeutic effects with fewer adverse effects, compared to western medicine alone [9]. Traditional herbal medicines contain multiple ingredients which target several pathways. The mechanism of action of the medicine is often complex and, in some cases, unclear. Therefore, investigating their efficacy through conventional trials is often difficult [10]. Network pharmacology is a novel approach that can be used to study the advantages of traditional herbal medicines. This approach can investigate the underlying mechanisms of multiple ingredients of herbal medicine by analyzing the compound-target network, protein–protein interaction network, as well as related signaling pathways. A network pharmacology study published recently indicated eight main bioactive compounds of *Curcumae Radix* with 45 targets related to depression and predicted the signaling pathways for the antidepressant effect of this medicinal herb [11].

*Atractylodes lancea* (Thunb.) DC. (AL), also known as Cangzhu in traditional Chinese medicine, or Changchul in Korean medicine has been widely used for the treatment of various symptoms and diseases, including gastrointestinal diseases, rheumatic diseases, the common cold, and influenza. AL possesses anti-inflammatory, antipyretic, and anticancer activities and has shown therapeutic effects on the central nervous system [12]. AL forms a part of several traditional herbal prescriptions used for the treatment of depression, such as Xiao-yao-san and Yueju-Wan [13,14]. Natural compounds derived from AL, including β-sitosterol and atractylenolide I, have exhibited antidepressant effects in mouse models [15,16]. However, the underlying mechanism of action of AL in the treatment of depression has not been investigated to date.

In this study, network pharmacology and molecular docking were used to investigate the underlying mechanism of action of AL in the treatment of depression, and its antidepressant effect was verified using a mouse model (Figure 1). The active components, potential antidepressant targets, and signaling pathways of AL were analyzed. Molecular docking was also conducted to predict the interactions between active compounds and potential targets. Behavioral tests were performed for the validation of the antidepressant activity of AL. The current study provides evidence of the therapeutic potential of AL in the treatment of depression and could help expand its use in other clinical indications.

## 2. Materials and Methods

### 2.1. Screening for Active Components of Atractylodes lancea (Thunb.) DC. (AL)

The information about the components of AL was retrieved from the traditional Chinese medicine systems pharmacology database and analysis platform (TCMSP) (https://old.tcmsp-e.com/tcmsp.php, accessed on 16 June 2022) [17]. A literature search with the keyword “*Atractylodes lancea* (Thunb.) DC.” was performed using the Pubmed database (https://pubmed.ncbi.nlm.nih.gov/, accessed on 16 June 2022). The criteria for active component screening included molecular weight (MW) from 180 to 500 Dalton, oral bioavailability (OB) ≥ 20%, drug-likeness (DL) ≥ 0.1, blood–brain barrier (BBB) ≥ −0.3, and Caco-2 permeability ≥ −0.4.

### 2.2. Screening for Antidepressant Targets of AL

The DisGeNET database (http://www.disgenet.org/, accessed on 16 June 2022) was used for identifying depression-related targets (Disease ID: C0011570) [18]. Targets with scores ≥ 0.3 were selected as potential targets for further analysis. The TCMSP database was used to access the targets of the active components of AL. The AL targets were compared with depression-related targets to select the common ones.

### 2.3. Compound-Target Network Construction

The network of the common targets between AL and depression was constructed and visualized using Cytoscape 3.8.2 (Cytoscape Consortium, San Diego, CA, USA) [19]. The nodes represent compounds and targets, while the edges represent the interaction between the compounds and their targets. Topological analysis was performed to obtain the degree, betweenness centrality, and closeness centrality of the nodes in the network. Betweenness centrality and closeness centrality indicate the ability of a node to control and spread information over a network. A node with a higher degree, betweenness centrality, and closeness centrality could play a more important role in the network.

### 2.4. Gene Ontology (GO) and Kyoto Encyclopedia Genes and Genomes (KEGG) Analysis

The GO function and KEGG pathway analysis of potential antidepressant targets of AL were performed using the Enrichr tool (https://maayanlab.cloud/Enrichr/, accessed on 24 June 2022) [20]. The top 10 GO biological processes, molecular functions, cellular component terms, and top 20 KEGG pathways were reported (*p*-value < 0.01). The diagrams of the KEGG pathways were retrieved from the KEGG PATHWAY Database (https://www.genome.jp/kegg/pathway.html, accessed on 24 June 2022).

### 2.5. Protein–Protein Interaction (PPI) Network Construction

PPI plays a crucial role in the regulation of biological processes and an investigation of PPI could help determine the potential therapeutic targets for the treatment of diseases [21]. In this study, the overlapping targets of AL compounds and depression were constructed using the Search Tool for the Retrieval of Interacting Genes/Proteins (STRING) (https://string-db.org/, accessed on 29 June 2022) for multiple proteins in humans (*Homo sapiens*) [22]. The minimum required interaction score was set at 0.4 (medium confidence). The PPI network and PPI enrichment *p*-values were retrieved.

### 2.6. Molecular Docking Analysis

The interaction between the AL compounds and target proteins was estimated using AutoDock Vina [23]. For docking analysis, the three-dimensional (3D) structure of the target proteins with the protein data bank (PDB) format was downloaded from (https://www.rcsb.org/, accessed on 29 June 2022) [24] and the 3D structures of the AL compounds were acquired from the PubChem database (https://pubchem.ncbi.nlm.nih.gov/, accessed on 29 June 2022). Docking scores (binding affinities) and binding modes between compounds and target proteins were automatically identified by AutoDock and PyMOL software [25]. The 2D diagrams of compound-target interaction were visualized using Discovery Studio Visualizer v21.1 program [26].

### 2.7. Preparation of AL Extract

AL was purchased from Omniherb (Daegu, South Korea). AL (45 g) was extracted with 30% ethanol at 80 °C for 3 h. The extract was filtered through filter paper (Whatman International, Maidstone, UK), evaporated using a vacuum evaporator, and freeze dried (yield 13.75% *w*/*w*).

### 2.8. Animal Experiment

ICR mice (male, 5 weeks) were obtained from Koatech (Gyeonggi, South Korea) and were allowed to acclimatize for seven days before the experiments. All animal experiments were approved by the Institutional Animal Care and Use Committee (IACUC) of Dongguk University, South Korea (Approval no. IACUC-2021-15). The mice were divided into four groups (*n* = 8 per group): the CON group (mice treated with the vehicle), the AL10 group (mice treated with 10 mg/kg of AL), the AL20 group (mice treated with 20 mg/kg of AL), and the MEM group (mice treated with memantine). AL was dissolved in 30% ethanol and administered intranasally (10 μL/mouse). Memantine (M9292, Sigma-Aldrich, St. Louis, MI, USA) was dissolved in saline and administered through the intraperitoneal route (3 mg/kg) as previously described [27]. All treatments were performed 30 min before the behavioral tests.

### 2.9. Tail Suspension Test (TST)

The TST was conducted to examine the antidepressant effect of AL in mice as previously described [28]. All mice were habituated to the test room 2 h before the experiment. The mouse was hung on a bar using adhesive tape attached to the tail. The total immobility time (s) of each mouse was recorded using the SMART v3.0 video tracking system (Panlab, Barcelona, Spain) during a 6 min session.

### 2.10. Social Interaction Test

The effect of AL on social behavior was investigated by the social interaction test as previously described with some modifications [29]. All mice were habituated to the test room 2 h before the experiment. Two mice from unfamiliar cages with the same treatment conditions were placed in the opposite corners of a box (30 cm × 30 cm × 30 cm). The total contact number was recorded in a 5 min session with the SMART v3.0 video tracking system (Panlab, Barcelona, Spain).

### 2.11. Open Field Test (OFT)

The effect of AL on exploratory and locomotor activity was evaluated using an OFT. All mice were habituated to the test room 2 h before the experiment. Each mouse was placed in the center of an OFT box (30 cm × 30 cm × 30 cm) and was allowed to explore freely for 5 min. The total distances (cm) covered and time spent in the center (s) by each mouse were assessed by the Smart V3.0 video tracking system (Panlab, Barcelona, Spain).

### 2.12. Statistical Analysis

Statistical analysis was performed using GraphPad Prism 9.0 (GraphPad Software, San Diego, CA, USA). The results represent means ± standard deviation (SD). Differences between groups were evaluated using the one-way ANOVA test and a *p*-value < 0.05 was considered statistically significant. All experiments were conducted at least three times independently.

## 3. Results

### 3.1. Screening for Active Components of AL

Based on the TCMSP database and literature search, 51 compounds in AL were identified (Appendix A). To identify the active components of AL, several TCMSP parameters, including molecular weight (MW), human oral bioavailability (OB), Caco-2 permeability (Caco-2), drug-likeness (DL), and blood–brain barrier (BBB), were used for screening. Among the 51 components of AL, 23 potentially active compounds were selected according to pharmacokinetic criteria, including 180 ≤ MW ≤ 500, OB ≥ 20%, Caco-2 ≥ −0.4, DL ≥ 0.1, and BBB ≥ −0.3, as shown in Table 1.

### 3.2. Potential Antidepressant Targets of AL and Network Analysis

Among the 23 potentially active components of AL, only 15 compounds had retrievable targets in the TCMSP database. These compounds targeted a total of 90 target genes (Appendix A). Depression-related targets were accessed from the DisGeNET database (Disease ID: C0011570). A total of 260 targets with a DisGeNET score ≥ 0.3 were selected (Appendix A). A total of 28 overlapping targets between AL and depression were considered the potential antidepressant targets of AL (Figure 2A, Table 2). Cytoscape 3.8.2 was used to visualize the compound-common target network. As shown in Figure 2B, the network contained 38 nodes (10 compounds and 28 targets) and 54 edges. The nodes represented common targets and compounds, and the edges represented the interaction between the compounds and common targets. The topological analysis of the main active components of AL is shown in Table 3.

### 3.3. GO and KEGG Enrichment Analysis

The GO enrichment analysis of the potential genes of AL of interest in depression treatment is shown in Figure 3A. These included biological processes, molecular functions, and cellular components. The GO biological process terms mainly included regulation of peptide hormone secretion, phospholipase C-activating G protein-coupled receptor signaling pathway, and positive regulation of acute inflammatory response. GO molecular function terms were mostly related to sodium:chloride symporter activity, monoamine transmembrane transporter activity, and transmitter-gated ion channel activity involved in the regulation of postsynaptic membrane potential. GO cellular component terms mainly comprised neuron projection, the integral component of the plasma membrane, and dendrite. The KEGG analysis revealed the pathways related to the antidepressant potential of AL, including neuroactive ligand–receptor interaction, calcium signaling pathway, dopaminergic synapse, IL-17 signaling pathway, and pathways of neurodegeneration (Figure 3B). Figure 4 shows the neuroactive ligand–receptor interaction and calcium signaling pathways with the highlighted potential targets of AL. The list of genes related to the GO terms and KEGG pathways is shown in Appendix A.

### 3.4. PPI Network Construction

Figure 5A shows the PPI network of potential antidepressant targets of AL, including 28 nodes and 95 edges. The PPI enrichment *p*-value was <10^−16^ and the average node degree was 6.79. The node degrees of IL6, nitric oxide synthase 3 (NOS3), solute carrier family 6 member 4 (SLC6A4), estrogen receptor (ESR1), and tumor necrosis factor (TNF) were greater than 10, suggesting that these targets could play important roles in the effects of AL on depression (Figure 5B).

### 3.5. Molecular Docking Analysis

Molecular docking analysis was conducted to assess the binding affinities between the active components of AL and the proteins with the top 5 node degrees in the PPI network (IL6, NOS3, SCL6A4, ESR1, and TNF). The more negative docking scores could indicate more stable and stronger bindings between compounds and proteins [30]. The docking scores of the AL compounds and corresponding protein targets were negative and less than −6.0 kcal/mol, suggesting the strong binding affinity between them (Table 4 and Appendix A). The compound-protein binding modes are shown in Figure 6.

### 3.6. Validation of the Antidepressant Effect of AL in Mice

The antidepressant effect of AL was validated by behavioral testing in mice. Figure 7A indicates that the intranasal treatment with AL significantly reduced the immobility time in the TST (*p* < 0.05), compared to the CON group. Impaired social functioning was indicated as a typical aspect of depression and social interaction showed preventive effects on the development of depression-like symptoms in mice [31,32]. The results of the social interaction test demonstrated that AL treatment significantly increased the total contact number between two mice (*p* < 0.05) (Figure 7B). The locomotor and exploratory activity of the mice was assessed using the OFT. As shown in Figure 7C, AL did not show any statistically significant effects on the total distance as well as time spent in the center during the 10 min OFT session.

## 4. Discussion

Since 60 years ago, a decline in serotonin activity and concentration has been suggested as a pathophysiological mechanism of depression, and current antidepressants were developed based on this hypothesis [33]. Nevertheless, recent systematic reviews have found a weak correlation between serotonin and depression, necessitating further research into alternate pathophysiological mechanisms and the development of therapies based on them [34]. According to the TCM theory, liver dysfunction is considered one of the causes of depression, consequently leading to liver qi stagnation and spleen deficiency syndromes [35]. In addition, spleen deficiency can result in dampness that might lead to liver qi stagnation [36]. Earlier studies have shown that AL can strengthen the spleen and eliminate dampness, suggesting that AL shows potential in the treatment of depression. Therefore, this study was conducted to investigate the antidepressant mechanisms of AL using network pharmacology and the molecular docking approach. We found that AL is composed of several bioactive ingredients which could exert antidepressant effects by targeting multiple genes and pathways.

In the present study, we showed that 10 bioactive compounds of AL potentially targeted 28 genes related to depression. These 10 components are mainly terpenoids and sterols, including stigmasterol, 3β-acetoxyatractylone, wogonin, β-sitosterol, selina-4(14),7(11)-dien-8-one, atractylenolide I, atractylenolide II, atractylenolide III, patchoulene, and cyperene. The pharmacokinetic characteristics of these compounds demonstrated their strong ability to penetrate the BBB, suggesting that the antidepressant effects of AL may be through direct effects on the brain. One study demonstrated the antidepressant effect of stigmasterol via interaction with the N-methyl-D-aspartate (NMDA) receptors [37]. β-sitosterol showed effects on depression by decreasing the immobility time in the tail suspension test in mice [16]. Atractylenolide I and atractylenolide III alleviated depression-like behavior in rodent models of depression [15,38]. These results indicate the potential of AL, a herb containing these compounds in the treatment of depression.

The GO enrichment analysis revealed the potential mechanisms of action of AL on depression. In GO biological process terms, AL compounds mainly targeted peptide hormone secretion, phospholipase C-activating G protein-coupled receptor signaling pathway, and the regulation of inflammatory response. Several peptide hormones, such as adrenocorticotropic hormone, oxytocin, and vasopressin play important roles in the pathogenesis of depression [39,40,41]. A variety of G protein-coupled receptors (GPCRs) are activated by neurotransmitters, such as the serotonin, dopamine, and gamma-aminobutyric acid (GABA) receptors, and the disruption of the GPCR signaling pathways might lead to behavioral changes in psychiatric disorders, including depression [42]. Inflammatory response with elevated levels of inflammatory cytokines has been demonstrated to be associated with depression and antidepressant drugs could reduce inflammation [43]. In GO cellular component terms, neuron projection, the integral component of the plasma membranes, and dendrites were major sites of potential targets of AL, suggesting AL might directly target neural cells to exert its antidepressant effect. In GO molecular function terms, the potential targets were mainly involved in sodium:chloride symporter activity, monoamine transmembrane transporter activity, and transmitter-gated ion channel activity which play a role in the regulation of postsynaptic membrane potential. Serotonin (5-HT) transporters (SERTs), which are members of the sodium-chloride cotransporter gene family, play an important role in the action of antidepressant drugs [44]. Monoamine transporters located on the plasma membrane contribute to 5-HT regulation in the treatment of depression [45]. Many ion channels are involved in the polarization state of the neuronal membrane potential that affects the release of neurotransmitters and influences depressive behavior [46].

The KEGG enrichment analysis revealed that the AL compounds targeted multiple signaling pathways related to depression, which primarily include neuroactive ligand–receptor interaction, the calcium signaling pathway, dopaminergic synapse, the IL-17 signaling pathway, and the pathways of neurodegeneration. The neuroactive ligand–receptor interaction signaling pathway mainly includes neurotransmitter receptors, which play a crucial role in the pathophysiology of psychiatric diseases, including depression [47]. Our study indicated that AL targeted several neuroreceptors, such as the acetylcholine receptor (CHRM), epinephrine/norepinephrine receptor (ADR), dopamine receptor (DRD), 5-HT receptor (HTR), opioid receptor (OPR), and GABA receptor (GABR), suggesting that the antidepressant effect of AL might be through regulating these receptor activities. A previous study has demonstrated the role of the calcium signaling pathway in depression and the disruption of the calcium-calmodulin-NOS-guanylyl cyclase signaling pathway could result in antidepressant effects [48]. Here, AL compounds targeted key proteins in calcium signaling, including G protein-coupled receptors (GPCRs), vascular endothelial growth factor-A (VEGF-A), calmodulin (CALM), and NOS, thus demonstrating the antidepressant potential of AL. The pathways of inflammation and neurodegeneration are important in the development of depression [49]. T helper 17 (Th17) cells and their effector cytokine IL-17 stimulate the activation of microglial cells, neuroinflammatory response, and neuronal damage, which are associated with depressive behavior [50]. These findings suggest that AL exerts its antidepressant effect by regulating multiple signaling pathways.

The PPI network analysis showed that IL6, NOS3, SLC6A4, ESR1, and TNF had the highest degrees in the network, suggesting that these proteins might play more important roles in the antidepressant effect of AL. In addition, molecular docking analysis demonstrated strong bindings between these potential proteins and corresponding compounds. TNF-α and IL-6 are two common inflammatory cytokines involved in the neuroinflammatory response in depression [51]. The levels of TNF-α and IL-6 were positively correlated with the severity of depression [52]. Fluoxetine, the main medication for the treatment of depression reduces the levels of IL-6 and TNF-α in patients [53]. NOS3 (or endothelial NOS) regulates the synthesis of NO, which plays an important role in the pathogenesis of depression by modulating neurotransmitter systems, and blocking NOS activity might result in antidepressant effects [54]. SLC6A4 is a serotonin transporter regulating serotonergic neurotransmission and is associated with depression [55]. ESR1 is an estrogen receptor that might be involved in the etiology of postpartum depression by regulating 5-hydroxytryptamine (5-HT) signaling [56]. These results imply that IL6, NOS3, SLC6A4, ESR1, and TNF proteins may be the key targets of the active compounds of AL in the treatment of depression.

The antidepressant activity of AL was validated by intranasal administration in the mice model and subsequent measurements of behavioral outcomes. Accumulating evidence has demonstrated that compared with other administration routes, such as oral administration or injection, intranasal administration of antidepressant drugs offers many advantages, including the fast onset of action, lower doses, fewer adverse effects, avoiding first-pass metabolism in the gastrointestinal tract, and liver, and increase in the bioavailability [57]. In the current study, the mice were treated with intranasal AL for 30 min before behavioral testing. TST is one of the most common behavioral tests to screen potential antidepressant agents [28]. In the TST, the mice were placed in an inescapable situation by suspending their tail with adhesive tapes. In response to this short-term stress, the mice develop an immobile behavior. Antidepressant drugs reduce immobility and increase escape activity [58]. In this study, we showed that AL treatment significantly decreased the immobility time in the TST, suggesting that AL could have an acute antidepressant effect. Impairment of social functioning was observed in patients with depression and social interaction could prevent the symptoms of depression in mice [31,32]. We found that AL significantly increased the total contact number between two mice in the social interaction test, implying that AL might exert beneficial effects against depression by improving social functioning. Results of the OFT showed that AL did not alter the locomotor and exploratory activity of mice, suggesting that AL did not possess a therapeutic effect against anxiety-like behavior. In keeping with a previous study, memantine, the positive control drug, also exerted an acute antidepressant effect and not an anxiolytic effect in mice [27].

The present study had some limitations. Additional studies should be conducted for confirmation of the signaling pathways mediating the antidepressant effect of AL. Stress models, such as chronic social defeat stress or social isolation stress should be used to investigate the long-term antidepressant effect of AL in addition to the acute effects as shown in this study. Therefore, further studies will be needed to confirm the beneficial effects of AL on depression.

## 5. Conclusions

In conclusion, network pharmacology analysis indicated 10 bioactive compounds of AL with 28 potential targets in the treatment of depression. GO and KEGG enrichment analysis revealed that these targets could directly affect neural cells and exert their action mainly through the regulation of neuroinflammation and neurotransmitter signaling. In addition, molecular docking analysis showed strong binding affinities between antidepressant target proteins and corresponding compounds. The antidepressant effect of AL was validated in the animal model, as inferred from a reduction in immobility time in the TST and an increase in the total contact number in the social interaction test. These findings provide a basis for further studies as well as for the clinical application of AL in the treatment of depression.

## Figures and Tables

**Figure 1 life-12-01925-f001:**
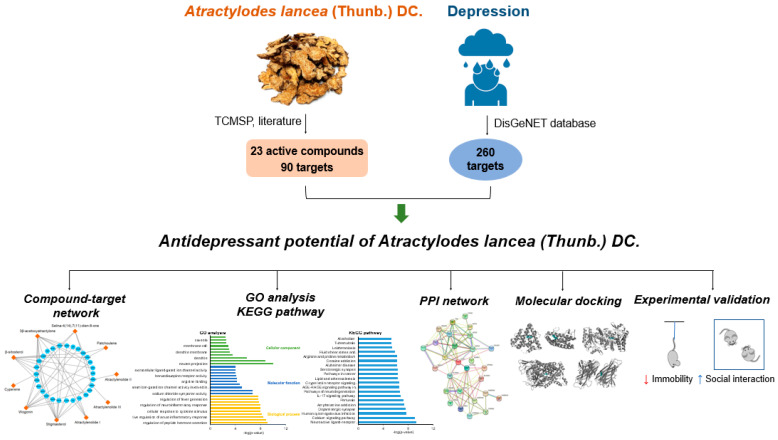
Workflow of the network pharmacological-based study. TCSMP: Traditional Chinese medicine systems pharmacology database and analysis platform; GO: Gene Ontology; KEGG: Kyoto Encyclopedia of Genes and Genomes; PPI: Protein–Protein Interaction Network; ↓: decrease; ↑: increase.

**Figure 2 life-12-01925-f002:**
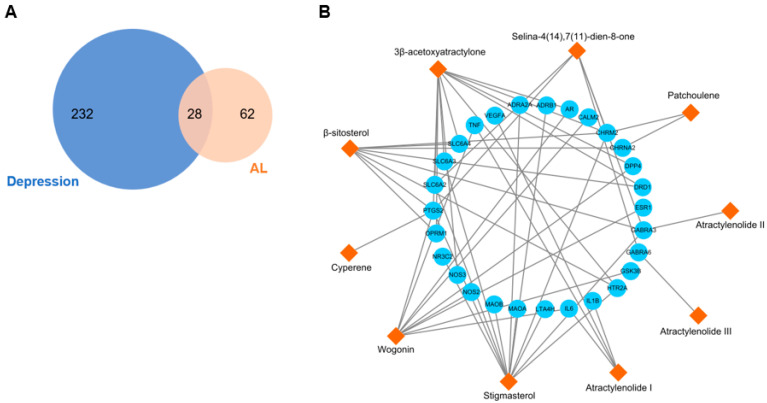
Network of the common targets between AL compounds and depression. (**A**) A Venn diagram shows the intersection between AL targets and depression-related targets. (**B**) Compound-target network of the overlapping targets between AL and depression. The orange diamonds represent the compounds, and the blue circles represent the targets. The edges represent the interaction between compounds and targets.

**Figure 3 life-12-01925-f003:**
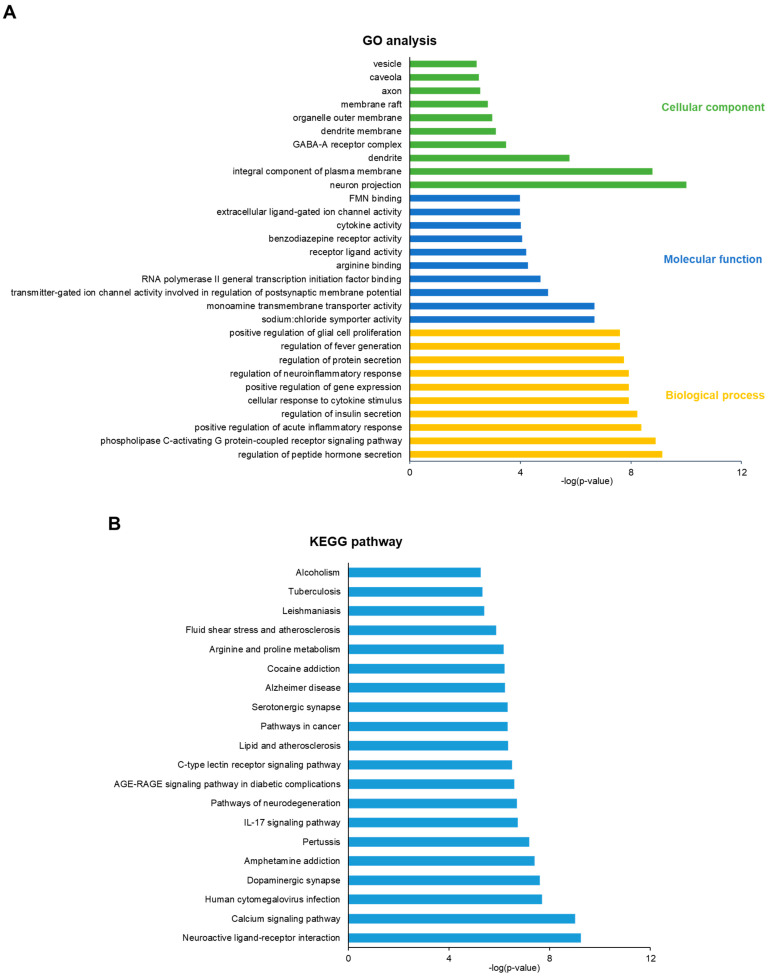
GO and KEGG analysis of the potential targets. (**A**) GO enrichment (Biological Process, Molecular Function, Cellular Component). (**B**) KEGG pathway analysis. GABA-A; γ-Aminobutyric acid sub-type A; FMN: flavin mononucleotide; AGE-RAGE: advanced glycation end products receptor for advanced glycation end products; IL: interleukin.

**Figure 4 life-12-01925-f004:**
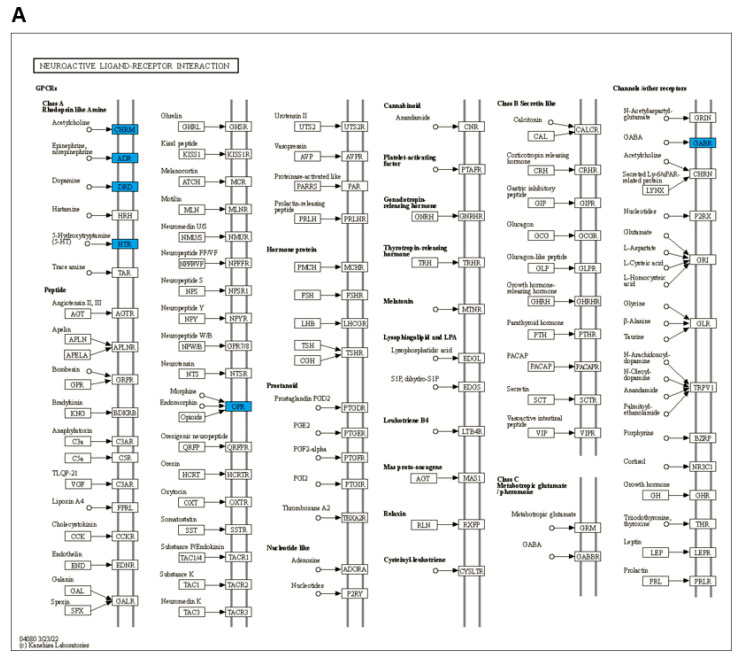
KEGG enrichment pathways. (**A**) Neuroactive ligand–receptor interaction signaling pathway. (**B**) Calcium signaling pathway. The blue color indicates potential antidepressant targets of AL.

**Figure 5 life-12-01925-f005:**
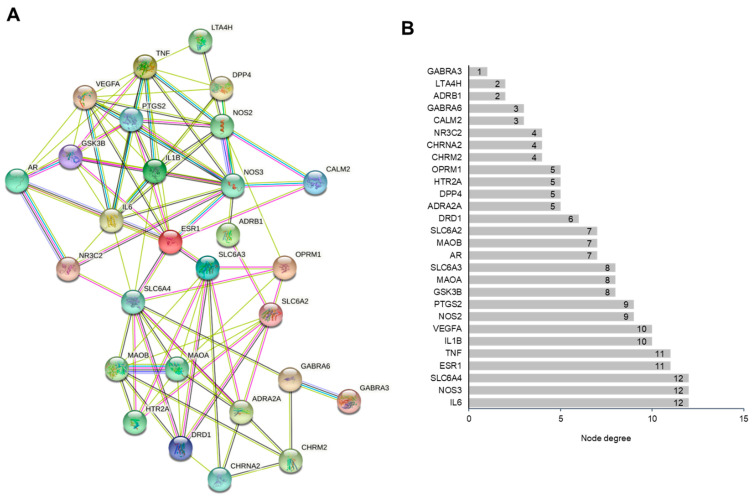
PPI network construction of the potential targets. (**A**) PPI network. The nodes represent proteins, and the edges represent the interaction between proteins. (**B**) Node degrees of proteins in the PPI network.

**Figure 6 life-12-01925-f006:**
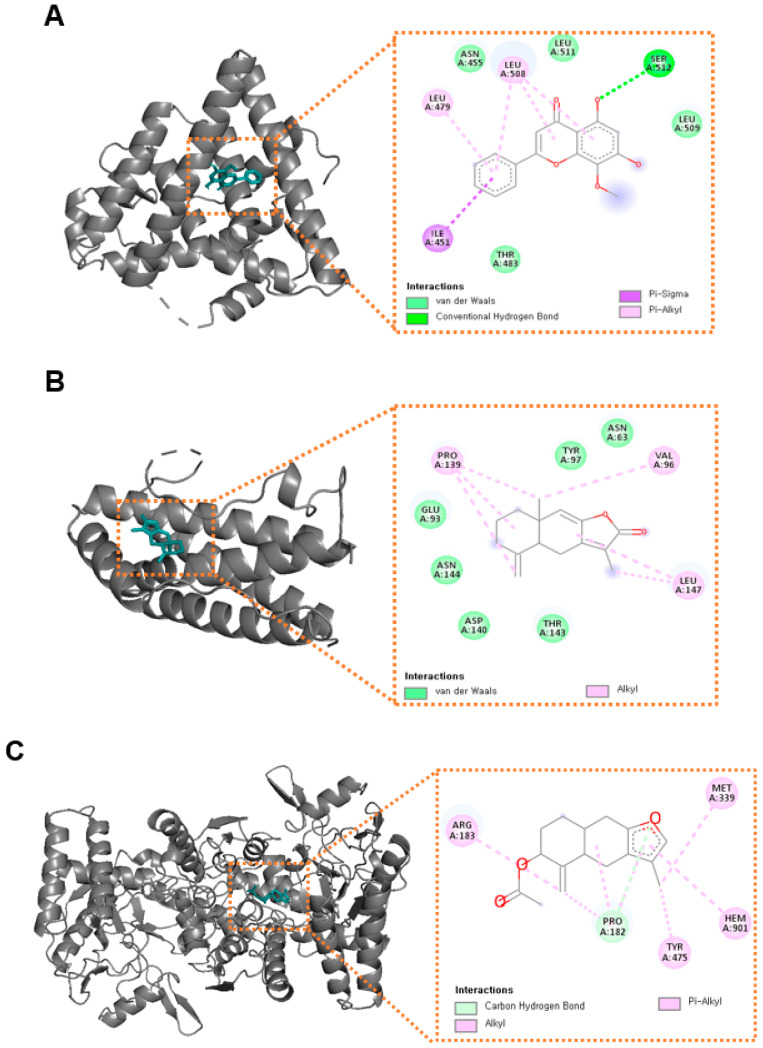
Molecular docking modes of compounds in AL and targets. (**A**) Wogonin with ESR1, (**B**) Atractylenolide I with IL-6, (**C**) 3β-acetoxyatractylone with NOS3, (**D**) 3β-acetoxyatractylone with SCL6A4, (**E**) Atractylenolide I with TNF. ESR1: estrogen receptor 1; IL6: interleukin 6; NOS3: nitric oxide synthase 3; SLC6A4: solute carrier family 6 member 4; TNF: tumor necrosis factor.

**Figure 7 life-12-01925-f007:**
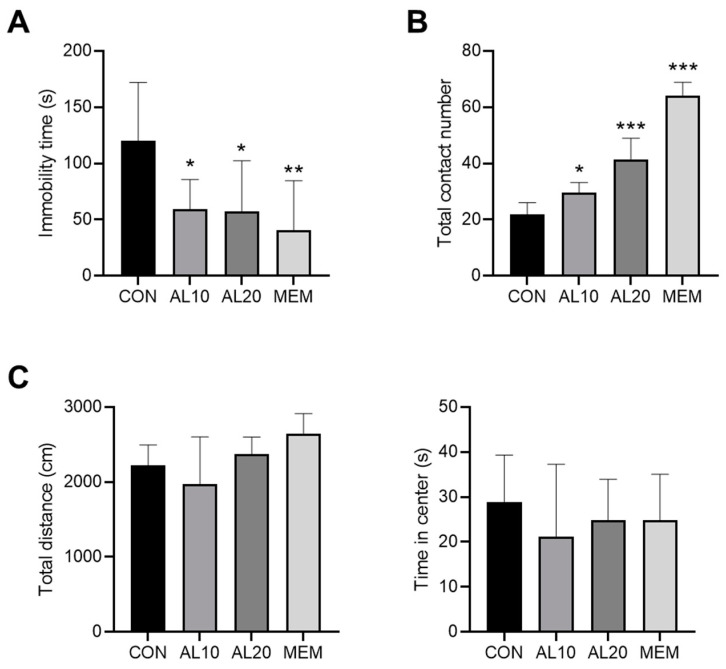
Validation of the antidepressant effect of AL in mice. (**A**) The immobility time in a 6 min session of TST was recorded. (**B**) The total contact number in a 5 min session of the social interaction test was recorded. (**C**) The total distance and time spent in the center in a 5 min OFT session were recorded. Data represent means ± SDs (*n* = 8 per group). * *p* < 0.05, ** *p* < 0.01, *** *p* < 0.001 vs. the CON group. CON: control; AL: *Atractylodes lancea* (Thunb.) DC.; MEM: memantine; TST: Tail suspension test; OFT: Open field test.

**Table 1 life-12-01925-t001:** Active components of *Atractylodes lancea* (Thunb.) DC.

Mol ID	Molecule Name	MW	OB (%)	Caco-2	BBB	DL
MOL000086	(24S)-5β-stigmastan-3β-ol	416.81	25.32	1.41	1.18	0.75
MOL000188	3β-acetoxyatractylone	274.39	40.57	1.22	1.04	0.22
MOL000167	3β-hydroxyatractylone	232.35	21.17	1.18	0.97	0.15
MOL000189	Acetyl atractylodinol	240.27	25.47	1.42	0.69	0.13
MOL000043	Atractylenolide I	230.33	37.37	1.3	1.29	0.15
MOL000044	Atractylenolide II	232.35	47.5	1.3	1.37	0.15
MOL000178	Atractylenolide III	248.35	31.66	0.75	0.64	0.17
MOL000164	Atractylone	216.35	33.91	1.74	1.83	0.13
MOL000187	Butenolide B	234.32	61	0.65	0.45	0.15
MOL000175	Cyperene	204.39	51.1	1.81	2.13	0.11
MOL000092	Daucosterin_qt	414.79	36.91	1.42	1.15	0.76
MOL000094	Daucosterol_qt	414.79	36.91	1.3	0.87	0.76
MOL000194	Patchoulene	204.39	51.71	1.8	2.21	0.11
MOL000060	Selina-4(14),7(11)-dien-8-one	218.37	32.31	1.42	1.57	0.1
MOL000184	Stigmastenone	412.77	39.25	1.42	1.22	0.76
MOL000449	Stigmasterol	412.77	43.83	1.44	1	0.76
MOL000186	Stigmasterol-3-O-β-D-glucopyranoside_qt	412.77	43.83	1.31	0.9	0.76
MOL000173	Wogonin	284.28	30.68	0.79	0.04	0.23
MOL000085	β-daucosterol_qt	414.79	36.91	1.3	0.88	0.75
MOL000032	β-eudesmol	222.41	26.09	1.32	1.38	0.1
MOL000358	β-sitosterol	414.79	36.91	1.32	0.99	0.75
MOL000088	β-sitosterol 3-O-glucoside_qt	414.79	36.91	1.3	0.91	0.75
MOL000095	Δ-7-stigmastenol	416.81	25.32	1.31	0.98	0.75

MW: molecular weight; OB: oral bioavailability; DL: drug-likeness; BBB: blood–brain barrier

**Table 2 life-12-01925-t002:** Potential anti-depressant targets of AL.

No.	Gene	Uniprot	Protein
1	*ADRA2A*	P08913	Alpha-2A adrenergic receptor
2	*ADRB1*	P08588	Beta-1 adrenergic receptor
3	*AR*	P10275	Androgen receptor
4	*CALM2*	P0DP23	Calmodulin 2
5	*CHRM2*	P08172	Muscarinic acetylcholine receptor M2
6	*CHRNA2*	Q15822	Neuronal acetylcholine receptor subunit alpha-2
7	*DPP4*	P27487	Dipeptidyl peptidase IV
8	*DRD1*	P21728	Dopamine D1 receptor
9	*ESR1*	P03372	Estrogen receptor
10	*GABRA3*	P34903	Gamma-aminobutyric-acid receptor alpha-3 subunit
11	*GABRA6*	Q16445	Gamma-aminobutyric-acid receptor alpha-6 subunit
12	*GSK3B*	P49841	Glycogen synthase kinase-3 beta
13	*HTR2A*	P28223	5-hydroxytryptamine 2A receptor
14	*IL1B*	P01584	Interleukin-1 beta
15	*IL6*	P05231	Interleukin-6
16	*LTA4H*	P09960	Leukotriene A-4 hydrolase
17	*MAOA*	P21397	Amine oxidase [flavin-containing] A
18	*MAOB*	P27338	Amine oxidase [flavin-containing] B
19	*NOS2*	P35228	Nitric oxide synthase, inducible
20	*NOS3*	P29474	Nitric-oxide synthase, endothelial
21	*NR3C2*	P08235	Mineralocorticoid receptor
22	*OPRM1*	P35372	Mu-type opioid receptor
23	*PTGS2*	P35354	Prostaglandin G/H synthase 2
24	*SLC6A2*	P23975	Sodium-dependent noradrenaline transporter
25	*SLC6A3*	Q01959	Sodium-dependent dopamine transporter
26	*SLC6A4*	P31645	Sodium-dependent serotonin transporter
27	*TNF*	P01375	Tumor necrosis factor
28	*VEGFA*	P15692	Vascular endothelial growth factor A

**Table 3 life-12-01925-t003:** Topological analysis of AL compounds in the compound-target network.

Compound	Closeness Centrality	Betweenness Centrality	Degree
Stigmasterol	0.357262341	0.430232558	12
3β-acetoxyatractylone	0.256199505	0.411111111	11
Wogonin	0.429541937	0.411111111	9
β-sitosterol	0.138956867	0.393617021	8
Selina-4(14),7(11)-dien-8-one	0.134796286	0.37755102	5
Atractylenolide I	0.107357357	0.246666667	4
Patchoulene	0.005315137	0.284615385	2
Cyperene	0	0.330357143	1
Atractylenolide III	0	0.220238095	1
Atractylenolide II	0	0.264285714	1

**Table 4 life-12-01925-t004:** Docking scores of compounds in AL and potential targets.

	Docking Score (kcal/mol)				
Compound	ESR1	IL6	NOS3	SLC6A4	TNF
3β-acetoxyatractylone	-	-	−7.8	−7.2	-
Atractylenolide I	-	−6.6	-	-	−7.7
Wogonin	−6.7	−6.3	-	-	−6.4
β-sitosterol	-	-	-	−6.8	-

## Data Availability

The data presented in this study are available in this article.

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
