# Peer review of "Network Pharmacology and Experimental Validation to Investigate the Antidepressant Potential of Atractylodes lancea (Thunb.) DC."

_life, 2022, doi:10.3390/life12111925_

Round 1
Reviewer 1 Report
Article
Network pharmacology-based study on the antidepressant potential of Atractylodes lancea (Thunb.) DC.
This study demonstrated the antidepressant potential of Atractylodes lancea (Thunb.) DC. (AL), which provides evidence for pursuing further studies to develop a novel antidepressant. Network pharmacology and molecular docking were used and antidepressant effect was verified using a mouse model.
As an outcome, AL moieties that could serve as putative drug candidates for control and therapy of depression, are identified.
With some limitations (short-term study and presentation of only acute antidepressant effect on mouse model), this study is good designed. The work is interested and can be accepted for publication in the Life, after minor revision addressing the following points.
Introduction: Does other similar studies using network pharmacology and molecular docking of this or other bioactive plant components in humans (not only mouse model) exist in the literature? Compare the findings.
Results:
- Could Figure 2B (central part) presentation be simpler? Its important figure for the results presentation, but can’t be visualized properly in this form. Please correct and use more suitable font.
Author Response
[Comment 1] Introduction: Does other similar studies using network pharmacology and molecular docking of this or other bioactive plant components in humans (not only mouse model) exist in the literature? Compare the findings.
Response: Thank you for this question. Our study performed network pharmacology and molecular docking using genes and proteins from humans, not mice. It should be noted that the DisGeNET database integrates information of human gene-disease associations and variant-disease associations (https://www.disgenet.org/dbinfo). In addition, as described in section 2.5, “The overlapping targets of AL compounds and depression were constructed using the Search Tool for the Retrieval of Interacting Genes/Proteins (STRING) (https://string-db.org/) for multiple proteins in humans (Homo sapiens)”.
[Comment 2] Results: Could Figure 2B (central part) presentation be simpler? It’s important figure for the results presentation but can’t be visualized properly in this form. Please correct and use more suitable font.
Response: Thank you for this suggestion. We corrected Figure 2B to visualize the results more properly.
Reviewer 2 Report
1.Title of the manuscript can be modified to make it more attractive for readers
2. Entire manuscript should be revised again to remove grammatical errors and to improve English language
3. At the start of manuscript, line number 9, the heading simple summary should look awkward. It should be either highlights or no need of it
4. On line number 21, it should be network pharmacology, molecular docking and animal studies instead of only network pharmacology and molecular docking.
5. On line 37 it should be contact hour instead of contact number.” increasing the total contact number in the social interaction test”
6. On line number 153 “AL was dissolved in 152 30% ethanol and administered intranasally (10 μl/mouse).” Don’t you think 30% alcohol itself will produce effect on CNS. Please add some reference here if you have.
7. Why you are giving AL intranasally and memantine intraperitoneally?
8. On page number 14, from docking results we cannot clearly see the interactions among compound and residue. Either add 2D images or discuss the interactions in result section.
9. Please provide your docking results for all compounds in supplementary data. You have 10 active compounds but docking is for five only. I know they showed best results but if you will include all docking results in supplementary data it would be more informative
10. On page number 15, the graphical representation, in graph C and D you did not added steric marks to indicate significance of results.
11. On line number 290, you mentioned “depression is caused by liver dysfunction, consequently leading to liver qi stagnation and spleen deficiency syndromes” This shows depression is only caused by liver dysfunctioning. Please rectify this
Author Response
[Comment 1] Title of the manuscript can be modified to make it more attractive for readers.
Response: Thank you for your suggestion. We changed the title of the manuscript to “Network pharmacology and experimental validation to investigate the antidepressant potential of Atractylodes lancea (Thunb.) DC.”.
[Comment 2] Entire manuscript should be revised again to remove grammatical errors and to improve English language.
Response: Thank you for your suggestion. We have carefully checked and revised some grammatical errors. And the entire manuscript was corrected by a professional English editing institution.
[Comment 3] At the start of manuscript, line number 9, the heading simple summary should look awkward. It should be either highlights or no need of it.
Response: Thank you for your comment. However, this heading has followed the template of the Life journal and we should keep it accordingly.
[Comment 4] On line number 21, it should be network pharmacology, molecular docking and animal studies instead of only network pharmacology and molecular docking.
Response: We appreciated your advice. We revised the sentence to make this point clear.
[Comment 5] On line 37 it should be contact hour instead of contact number.” increasing the total contact number in the social interaction test”.
Response: Thank you for showing this point. In the social interaction test, in addition to the contact duration, the total contact number can also be a criterion for assessing the social deficits improvement, a characteristic of the antidepressant effect (Mol Neurobiol. 2017, 54(10):8152-8161; Behav Brain Res. 2007, 180(1):69-76; Biofactors. 2020, 46(1):38-54).
[Comment 6] On line number 153 “AL was dissolved in 30% ethanol and administered intranasally (10 μl/mouse).” Don’t you think 30% alcohol itself will produce effect on CNS. Please add some reference here if you have.
Response: We thank the reviewer for bringing up this point. We used 30% ethanol as the vehicle following a previous study that also dissolved drugs in ethanol for intranasal administration to investigate the antidepressant effect of the drugs (Evid Based Complement Alternat Med. 2011, 2011:512697). Moreover, only a small volume (10 μl) of 30% ethanol was used and the CON group was also treated with the same vehicle to establish a baseline for determining the effect of AL.
[Comment 7] Why you are giving AL intranasally and memantine intraperitoneally?
Response: Intranasal administration of AL should be considered as one novelty of our study, and to the best of our knowledge, this is the first paper that reports the effects of AL via the intranasal route. We also mentioned in the manuscript that “compared with other administration routes, such as oral administration or injection, intranasal administration of antidepressant drugs offers many advantages, including the fast onset of action, lower doses, fewer adverse effects, avoiding first-pass metabolism in the gastrointestinal tract, and liver, and increase in the bioavailability.”
In this study, we would like to check if AL could exert a fast-acting antidepressant effect in mice, and the positive control (memantine) should also exert the same effect within the same time course to be compared. Based on a previous study, memantine administered intraperitoneally 30 min before the behavioral tests showed rapid-acting antidepressant effects (Neurosci Lett. 2006, 395(2):93-97), which is appropriate for our aim of the experiment. Therefore, we followed the conditions of using memantine from that study.
[Comment 8] On page number 14, from docking results we cannot clearly see the interactions among compound and residue. Either add 2D images or discuss the interactions in result section.
Response: Thank you very much for your suggestion. We added the 2D diagrams of compound-target interactions in Figure 6. Since CB Dock cannot generate the 2D interaction between compounds and targets, we used different programs (AutoDock, PyMOL, and Discovery Studio Visualizer) for the molecular docking analysis and 3D/2D visualization as indicated in section 2.6.
[Comment 9] Please provide your docking results for all compounds in supplementary data. You have 10 active compounds but docking is for five only. I know they showed best results but if you will include all docking results in supplementary data it would be more informative.
Response: Thank you very much for your suggestion. We included the docking results of other compounds with their potential targets in the supplementary Table S7.
[Comment 10] On page number 15, the graphical representation, in graph C and D you did not added steric marks to indicate significance of results.
Response: Thank you for your comment. As described in the Results part, there were no statistically significant differences between groups in the total distance and time spent in the center in the open field test (OFT).
[Comment 11] On line number 290, you mentioned “depression is caused by liver dysfunction, consequently leading to liver qi stagnation and spleen deficiency syndromes” This shows depression is only caused by liver dysfunctioning. Please rectify this.
Response: Thank you very much for your suggestion. We revised the sentence to rectify this.
Round 2
Reviewer 2 Report
Authors have incorporated all the raised concerns.